# Sociodemographic and Psychological Variables and Concerns Related to COVID-19 Vaccination among Polish Citizens

**DOI:** 10.3390/ijerph19159507

**Published:** 2022-08-02

**Authors:** Estera Twardowska-Staszek, Irmina Rostek, Krzysztof Biel

**Affiliations:** Faculty of Education Jesuit University Ignatianum in Krakow, Kopernika 26, 31-501 Kraków, Poland; irmina.rostek@ignatianum.edu.pl (I.R.); krzysztof.biel@ignatianum.edu.pl (K.B.)

**Keywords:** pandemic, COVID-19, vaccination, attitudes, concerns

## Abstract

The aim of this study performed during the fourth wave of the pandemic was to analyse differences in sociodemographic and psychological variables between those who cite concerns regarding COVID-19 vaccination and those who do not, as well as the differences between those whose concerns stem from a negative evaluation of vaccines and those whose concerns are based on a positive evaluation of vaccines. The study included 417 participants aged 18 to 76 years (M = 34, SD = 13.9). Among the respondents, 89% were female. A survey questionnaire on sociodemographic variables and standardized research tools were used: mood (UMACL), emotions (PANAS), satisfaction with life (SWLS), optimism (LOT-R), and coping with stress (CISS). The results of the study indicate that the elderly and working people are concerned about inadequate vaccination of the population, whereas students are concerned about the pressure of compulsory vaccination. People who are concerned about inadequate vaccination of population are more likely to experience concerns about various stressors. Our results do not indicate a relationship between psychological variables and vaccination-related concerns. The results obtained may be the basis for the identification of target groups in order to adapt social campaigns promoting vaccination against COVID-19 in Poland.

## 1. Introduction

### 1.1. The COVID-19 Pandemic and the Ways to Overcome It

Since December 2019, when the first case of coronavirus infection SARS-CoV-2 was detected in Wuhan, China, more than 6,387,770 people worldwide have died from COVID-19, including 116,470 in Poland (18 July 2022) [1].

Strict sanitary security measures were implemented in almost every country to limit the spread of the virus. However, it soon became clear that sanitation compliance, or even total lockdown, was not enough to contain the COVID-19 pandemic. Therefore, almost from the very beginning of the COVID-19 pandemic, intensive research was launched to find effective treatments and prevent the spread of the SARS-CoV-2 coronavirus [2,3]. The first treatments consisted of antiviral drugs and antibiotics [4], immunomodulatory drugs [5], anti-inflammatory drugs, steroids, antimalarial drugs [6] and even convalescent plasma [7]. Unfortunately, most of the proposed approaches were not sufficiently effective in the treatment of COVID-19. Therefore, vaccine research was conducted in parallel with research on treatments. Many people welcomed the development of a vaccine formulation with the hope of defeating the pandemic. On 21 December 2020 (almost a year after the so-called patient zero was identified), the European Medicines Agency (EMA) conditionally approved Europe’s first COVID-19 vaccine developed by Pfizer BioNTech. Other formulations were conditionally approved in the following weeks: Moderna, Astra Zeneca, and Johnson & Johnson [8,9,10,11]. As a result, vaccination against COVID-19 began in every EU Member State as well as worldwide.

### 1.2. COVID-19 Vaccination in POLAND

The first citizen of Poland was vaccinated on 27 December 2020, marking the beginning of the National Vaccination Programme against COVID-19. Mass vaccinations were planned out in several stages. Firstly, the programme covered health care workers, employees of social assistance homes, and municipal social assistance centres (stage 0). The next batches of vaccines were administered to residents of nursing homes, care and treatment centres, persons over 60 years of age, public services and forces, as well as teachers (stage I). Subsequently, vaccinations were administered to persons under 60 years of age with chronic diseases and persons directly ensuring the functioning of the basic activities of the state and exposed to a high risk of infection (stage II). In the last stage, the vaccines were made available to the remaining part of adult Polish citizens [12]. The government conducted numerous social campaigns to reassure citizens about the safety and efficacy of the vaccine, as well as to build trust and motivation. A government information platform was created where, in addition to information and answers to frequently asked questions, people could register for vaccination. A 24-h helpline and chatbot were launched. Numerous experts (virologists, epidemiologists) were invited to participate in the campaign. The main sources of the campaign were the media and the Internet.

However, no vaccination programme can be effective if people do not get vaccinated. Social acceptance seems to be the crucial factor. For vaccination to be effective in limiting the spread of the virus, a sufficient proportion of the population must be vaccinated. It is estimated that vaccinating 60–72% of the population can lead to so-called herd immunity, which significantly inhibits the spread of the virus [13]. However, it should be borne in mind that the SARS-CoV-2 coronavirus continues to mutate and it is very likely that the estimation will fail for novel coronavirus variants. In Poland, there are 22,479,540 fully vaccinated citizens [12].

### 1.3. Factors Important for the Acceptance/Rejection of the Idea of Vaccination against COVID-19

It has been shown that people have very different attitudes towards COVID-19 vaccination and these are informed by multifactorial determinants: demographic, cognitive, psychological, political, and cultural [14,15]. Understanding these factors has been one of the main areas of research interest in 2021 in almost every geographical area: in Poland [16,17,18,19], in Europe [20], in the United States [21,22], in Australia [23], and in other countries [24].

It is also worth emphasizing that changes in the level of acceptance of vaccines have been observed over time. A study by Oleksy and co. [25] shows that willingness to be vaccinated against COVID-19 continued to decrease with each successive wave of the pandemic: the highest willingness to get vaccinated was declared by respondents during the first wave and the lowest during the fourth wave.

#### 1.3.1. The Role of the Demographic Variables for the Acceptance/Rejection of the Idea of Vaccination against COVID-19

Researchers from many countries have looked for patterns to explain people’s attitudes towards vaccination. It turns out that the results obtained show major differences between countries [26,27]. The results obtained differed not only from country to country, but also depending on the methodology used in particular studies. Some results indicated that older (≥50) people in Poland, Germany, France, Sweden, the UK, and Canada were more likely to declare their willingness to receive the vaccine than younger respondents, but the opposite trend was observed in China. Women in Germany, France, Sweden, and Russia were significantly more likely to accept the vaccine than men in these countries. Those with higher education in Germany, France, the US, Ecuador, and India reported that they would accept the vaccine, but higher levels of education were associated with lower vaccine acceptance in Spain, the UK, and Canada [24].

Despite the differences, some patterns have in fact been identified. Results from a number of international studies indicated that demographic factors associated with a lower propensity to vaccinate were younger age, female gender, low education level, low income, and unemployment [28,29,30,31,32]. Proponents of vaccination, on the other hand, tend to be older, male, with higher levels of education and higher incomes [33,34]. The results of Polish studies mostly confirm this correlation (older age, male gender, a better self-assessed financial status were associated with greater willingness to be vaccinated against COVID-19) [18,19]. In contrast, other Polish studies revealed a greater willingness to get vaccinated in older people, women, people living in big cities, and people with higher education [16].

#### 1.3.2. The Role of the Cognitive Variables (Beliefs) for the Acceptance/Rejection of the Idea of Vaccination against COVID-19

A review of 82 studies [30] found that the most common factors influencing vaccine hesitancy include a perceived low efficacy of the vaccine, side effects, distrust towards the healthcare system, and information sources. Another important factor associated with the varying levels of vaccine acceptance is the perceived risk of becoming infected with COVID-19. If the perceived risk is high, this leads to increased acceptance of the vaccine [35]; if the perceived risk is low, this leads to a greater belief in conspiracy theories and lowers vaccine acceptance [29,31]. The proliferation of various conspiracy theories about COVID-19 has been regarded by the WHO as one of the greatest threats to effective pandemic control. Conspiracy theories, which lower trust in scientific expertise and government representatives, have caused some people to disregard sanitary safety rules and not take even a single dose of the COVID-19 vaccine. Interestingly, not all conspiracy theories have similar consequences. The effect depends on the content of a given conspiracy theory. A study by Oleksy and co. [19] shows that some may be associated with a decrease and some with an increase of motivation for health-seeking behaviour. Thus, the belief that COVID-19 is a biological weapon results in increased health-related activities, whereas the opposite effect can be observed in the case of pandemic denial or the belief that the pandemic is a means of restricting citizens’ rights and freedoms. Another study by Oleksy and co. [25] shows that the belief in conspiracy theories appears to be a considerable barrier preventing people from getting vaccinated.

#### 1.3.3. The Role of the Emotional Variables (Fears and Concerns) for the Acceptance/Rejection of the Idea of Vaccination against COVID-19

The above-mentioned beliefs are undoubtedly emotionally marked. The spectrum of emotions they carry includes both positive (e.g., hope and security) and negative emotions (e.g., fears and concerns).

The anxiety directly related to vaccination is connected with the concerns about the origin of the vaccine and about the vaccine being produced too quickly; in addition, some people consider the vaccine more dangerous than COVID-19 or question the threat of becoming infected with COVID-19 [36]. Poles, aside from expressing common concerns about the lack of adequate testing of the vaccine, the presence of side effects or the lack of efficacy in general, cited concerns related to the poor transport/storage of vaccines. However, over time, concerns about transport/storage diminished whereas concerns about lack of vaccine efficacy increased [16].

As the emotional component is an important part of the attitude towards vaccination, emotions are also a powerful tool used in campaigns to encourage vaccination. Campaigns appeal to both positive (related to hope for the future and referring to solidarity) and negative emotions (referring to the fear of the consequences of becoming infected). It is worth noting that the analyzes conducted in Australia showed greater effectiveness of campaigns referring to positive emotions than negative ones [37].

Therefore, our study focuses on the emotional aspect of attitudes towards vaccination, with particular emphasis on concerns about vaccination. We assume that, first of all, these concerns are varied. They may take the form of concerns about the consequences of inadequate vaccination of the population (we assume that such concerns occur in people who positively assess vaccination) and the form of concerns about the consequences of mandatory vaccination (we assume that such concerns occur in people who negatively assess vaccination). We also assume that in a certain part of the population none of these are present.

In addition, we assume that (paradoxically) admitting to vaccination-related concerns is easier (and therefore more variable) than admitting to behavior (vaccination versus non-vaccination), which may be a subject of strong social evaluation. Of course, we do not assume a direct translation of fears and concerns into behaviors, but we believe that the hypothesis that people who are concerned about insufficient vaccination of the population accept and are more likely to vaccinate is interesting and requires further verification.

### 1.4. The Aim of the Study

Vaccination aims to reduce the spread of the virus and alleviate symptoms if the illness is contracted. Given these health-related and social benefits, understanding the demographic and psychological variables associated with vaccine acceptance is crucial in the fight against the pandemic.

Therefore, we aim to analyse the differences in sociodemographic and psychological variables between those who cite fears related to vaccination and those who do not report such fears, as well as the differences between those whose fears result from a negative evaluation of the vaccines and those whose fears result from a positive evaluation. The results obtained may help to identify risk groups in order to adapt social campaigns promoting vaccination against COVID-19 in Poland.

## 2. Materials and Methods

### 2.1. Participants

The study involved 417 individuals aged 18 to 76 years (M = 34.38, SD = 13.9, median = 31, quartiles = 22–45). The criteria for the selection of study participants were the following: age (18 or older) and place of residence (Poland). The description of demographic variables is presented in Table 1.

### 2.2. Measures

The study was conducted using a survey questionnaire in Google form, which was shared through personal contacts and through social media using the snowball method. The questionnaire consisted of two parts, the first containing questions on sociodemographic variables and the second including 5 standardized research tools.

#### 2.2.1. Sociodemographic Variables

Demographic variables were collected using ad hoc-designed questions. The demographic variables examined were: age, gender, marital status, children/lack thereof, place of residence, education, employment. In addition, a question was asked about direct exposure to COVID-19 of the respondent or his/her relatives. Finally, the questionnaire required that the respondents indicate possible changes in economic status and name sources of stress resulting from the pandemic.

#### 2.2.2. Mood

The UMACL consists of 29 items, which are adjectives describing one’s mood. The respondent has to determine to what extent a given adjective describes his or her current mood using a 4-point scale: from “definitely” to “definitely not”. The overall mood score includes three mood dimensions: hedonic tone (HT, the subjective feeling of pleasure–unpleasure), tense arousal (TA, the subjective feeling of anxiety) and energetic arousal (EA, the feeling of energy to act) [38,39].

#### 2.2.3. Emotions

PANAS consists of 20 items, which are adjectives describing positive and negative emotions. The respondent evaluates the intensity in which these emotions occur in him/her using a 5-point scale: (1—“very slightly” or “not at all”, up to 5—“extremely”). As a result, we obtain scores on two subscales: positive affect (PA) and negative affect (NA) [40,41].

#### 2.2.4. Satisfaction with Life

The SWLS contains 5 statements relating to one’s own life. The respondent assesses to what extent he/she agrees with each statement using a 7-point scale (from 1—“strongly disagree” to 7—“strongly agree”). The total score indicates the degree of satisfaction with life (low, medium, high) [42,43].

#### 2.2.5. Optimism

The LOT-R contains 10 statements, and the participant assesses to what extent a given statement applies to him/her using a 5-point scale (from 0—“strongly disagree” to 4—“strongly agree”). The total score of the test indicates an overall life orientation (tendency towards pessimism, neutral orientation, tendency towards optimism) [42,44].

#### 2.2.6. Coping with Stress

The questionnaire contains 48 statements concerning various behaviours that people exhibit in stressful situations. The respondent answers on a 5-point Likert scale indicating the frequency with which a given activity is undertaken in stressful situations (from 1—“never” to 5—“very often”). The result of the questionnaire is presented in the form of three styles of coping in stressful situations: task-oriented coping (TOC), emotion-oriented coping (EOC), and avoidance-oriented coping (AOC). The last style may take the form of distraction (D) or social diversion (SD) [45,46].

### 2.3. Design and Procedure

The study was designed and conducted in accordance with the guidelines of the Declaration of Helsinki and approved by the Research Ethics Committee of the Ignatianum Academy in Kraków (on 15 June 2021). First, a survey questionnaire was developed in Google Forms. It consisted of two parts: the first included sociodemographic variables and the second contained standardised research tools. Then, the online survey questionnaire was distributed using the snowball method (through personal contacts and social media). Research participants were informed of the purpose of the study and assured that participation in the study was voluntary, anonymous and that they could opt out of completing and/or submitting their responses at any time. The survey was conducted in December 2021 and January 2022.

### 2.4. Data Analysis

Statistical analyses were performed in R, version 4.1.2. (R Core Team, Vienna, Austria) [47]. A comparison of qualitative variables in groups was performed using the chi-square test (with Yates correction for 2 × 2 tables) or Fisher’s exact test where low expected numbers appeared in the tables. A comparison of the values of quantitative variables in the two groups was performed using the Mann–Whitney test. A comparison of the values of quantitative variables in the three groups was performed using the Kruskal–Wallis test. When statistically significant differences were detected, post-hoc analysis was performed with Dunn’s test to identify statistically significantly different groups.

## 3. Results

In the course of the study, the respondents were asked to indicate factors that contributed to their stress in the current situation. Basing their answers on two questions, one referring to their concerns about insufficient vaccination of the population and the other referring to their resistance to vaccination mandates, allowed us to select three groups of participants. The first group were those who declared that they were concerned about inadequate vaccination of the population (Group A) *n* = 104 (24.94%), the second group were those who declared that they were concerned about the prospect of mandatory vaccination (Group B) *n* = 92 (22.06%), and the third group were those who reported no concerns about vaccination (Group C) *n* = 221 (53%).

Table 2 shows the differences between the distinguished groups in terms of sociodemographic variables. As the results indicate, only two sociodemographic variables point to a significant difference between the groups. Individuals who are concerned about the insufficiently vaccinated population were significantly older than the other two groups. In addition, the group had the highest percentage of working people. On the other hand, the lowest percentage of working people was found in the group which feared compulsory vaccination. The opposite is true when one examines the percentage breakdown in pupils and students.

Table 3 shows the differences between the groups in terms of contact with COVID-19, as well as economic situation and sources of stress during the pandemic. As the results indicate, the significant differences concern stressors alone.

Results indicate that the participants were most afraid of: the political situation in Poland (62.82%), impeded access to treatment for other diseases (61.63%), the national economic situation (56.59%), and the possibility of their loved ones contracting COVID-19 (49.88%). When comparing the results obtained by three groups, it was noted that in group A, respondents are most concerned about: the political situation in the country (80.77%), the national economic situation (69.23%), the possibility of their loved ones contracting COVID-19 (67.31%), and impeded access to treatment for other diseases (63.46%). The respondents in group B are most concerned about: impeded access to treatment for other diseases (65.22%), the national economic situation (57.61%), the political situation in the country (55.43%), and restrictions (50%). The subjects in group C are most concerned about: impeded access to treatment for other diseases (59.28%), the political situation in the country (57.47%), the national economic situation (50.23%), and the possibility of their loved ones contracting COVID-19 (47.51%).

Further analyses show that only two groups differ significantly in terms of some stressors: the concern about the possibility of contracting COVID-19 (by themselves and their relatives) was significantly greater in group A than in group B. People from group A were significantly more concerned about the political and economic situation in the country and about online teaching. On the other hand, people from group B were significantly more concerned about restrictions than people from group A.

Regarding the remaining variables, i.e., mood, emotions, life satisfaction, life orientation, and coping styles, there were no statistically significant differences between the studied groups. (Table 4).

## 4. Discussion

Whereas international and Polish studies to date mainly focus on finding the determinants of individual willingness or acceptance of vaccination, our study aimed to analyse vaccination fears. Thus, we focused on the search for an emotional component informing one’s attitude towards vaccination.

Our age-related findings showed that there is an association between age and concerns about vaccination in the general population. Older people are concerned about inadequate vaccination of the population and the resulting lack of herd immunity. Other Polish studies show that age is an important variable when it comes to vaccination willingness e.g., Babicki and Mastalerz-Migas [16], Raciborski and co. [17] and Oleksy and co. [25]. Since the beginning of the pandemic, the media have been reporting new cases worldwide, with an emphasis on the severity of disease symptoms and high mortality rates in the elderly. Young people could feel safe as they were reported to be either asymptomatic or mildly affected by the disease with symptoms resembling those of a common cold. Such media coverage increased fear of the disease in older people, whereas it was relatively low among young people. It should be added that the young, who did not fear for their own lives, did not notice that they were becoming a threat to the older members of society.

Our findings on gender did not show an association between gender and vaccination concerns in the population. Many studies that consider gender in terms of vaccination do not provide clear conclusions. Global reports show that men are generally more enthusiastic about vaccination than women [48]. Polish studies show contradictory trends and therefore are not conclusive on the matter. For example, Raciborski and co. [49] note that men show more willingness to vaccinate than women, whereas Babicki and Mastalerz-Migas [16] indicate the opposite.

Our study has found no association between place of residence and vaccination concerns in the population. Yet, a Polish study conducted by Raciborski and co. shows that people living in rural areas and cities with fewer than 100.000 inhabitants declare a lack of willingness to be vaccinated against COVID-19. According to the authors, this might be caused by the density of population, the degree of urbanisation and industrialisation, and the number of COVID-19 cases and COVID-19-related deaths reported in rural areas and smaller towns, which was lower there compared to big cities. Due to differences in COVID-19 pandemic dynamics between rural and urban areas, the proportion of respondents reporting fear of COVID-19 and a subsequent willingness to be vaccinated against COVID-19 in the countryside and small towns may be lower than among residents of big cities, where the risk of SARS-CoV-2 infection (e.g., on public transport or in a shop) appears to be higher [49].

Our study has found no association between education and vaccination concerns in the population. International and Polish studies are also inconclusive. Some reports indicate that a higher level of education correlates with vaccination acceptance [16,33,34]. Other studies do not show a clear correlation. A meta-analysis of studies from 19 countries with a high COVID-19 burden found that the effect of education level on confidence in COVID-19 vaccines varied across countries. People with higher education in Ecuador, France, Germany, India, and the USA reported that they would accept COVID-19 vaccine, in contrast to Canada, Spain, and the UK, where higher levels of education correlated with lower acceptance of COVID-19 vaccination [24].

Our results demonstrate an association between employment and concerns about vaccination levels of the population. Those in employment declared concerns about inadequate vaccination of the population. In contrast, those in education and study expressed fears related to vaccine mandates. Other studies confirm this observation and show that unemployed and low-income people were more likely to be vaccination opponents [28,29,30,31,32].

Our research conducted at the turn of 2021/2022 showed that the list of concerns experienced by Poles remained unchanged compared to the situation a year earlier. However, these concerns occur more frequently [50].

The differences in the stressors experienced by people from the two groups lead to further questions. Participants who were concerned about inadequate vaccination of the population reported experiencing stress significantly more often than people from groups concerned about the prospect of mandatory vaccination in the case of the five contexts we indicated. Yet, what is interesting, they were not in a worse psychological condition, whereas the logical consequence of a greater number of stressors would be a deterioration of, for example, mood. Perhaps the very attitude towards vaccination (we tentatively assume that people who are concerned about insufficient vaccination of the population accept and are more likely to vaccinate) is a protective factor. However, this is, of course, only a hypothesis that requires further verification.

Our results show that people who were concerned about citizens not getting vaccinated were more worried about their health and that of their loved ones in the context of COVID-19 infection. A similar conclusion was made by Patwary and co [35] based on a meta-analysis of studies from 33 countries—if the perceived risk of COVID-19 infection is high, it leads to an increased acceptance of the vaccine. If it is low, it leads to vaccine denial (the level of hesitancy will increase sixfold in people who are convinced that they will not be infected) [51]. This can be explained by the social-cognitive model proposed by Eberhardt and Ling [52]. According to the model, people are more likely to engage in protective behaviour when they believe that inaction poses a threat to them, and that protective behaviour reduces the threat. Thus, people who are afraid of getting sick will take protective action against getting sick, i.e., they will get vaccinated. The situation is different for people who will be more afraid of receiving the vaccine than of contracting the disease. In this case, inaction will constitute protective behaviour. There is also a possibility that the respondents concerned about compulsory vaccination simply doubt the existence of a pandemic, and in the case of such doubts, they do not see the sense of vaccination. Our survey also shows that those who express fears about insufficient vaccination of the population are more concerned about the political and economic situation in the country and about remote teaching, whereas the opponents of vaccination tend to worry more about current restrictions.

Our results do not indicate a relationship between psychological variables (mood, emotions, life satisfaction, optimism or coping styles) and vaccination-related concerns.

## 5. Conclusions

A large segment of the Polish population does not want to be vaccinated, which poses a risk of further spread of the virus. This attitude has many conditions and consequences. Therefore, a continuous monitoring of attitudes towards vaccination is an essential aspect of implementing appropriate strategies and achieving success.

However, it should be borne in mind that studying the willingness to be vaccinated is one thing and studying the emotional aspect of attitudes towards vaccination is another and seems to be a valuable lead to better understand the phenomenon. Our survey provides information that can be useful for the National COVID-19 Vaccination Programme architects. The efforts to promote vaccination against COVID-19 should target young people, learning, and studying. With this in mind, campaigns should be conducted in schools and universities and should be publicized through social media, i.e., in places where young people spend their time. In addition, vaccination campaigns should involve young, popular people with whom young people can identify.

It has been also demonstrated that people who are concerned about inadequate vaccination of the population are also concerned about the health of their loved ones. The altruistic motivation directed towards love, care, and responsibility seems to have been a considerable factor strengthening their willingness to get vaccinated [53]. Therefore, perhaps, instead of fighting conspiracy theories or threatening anti-vaccinationists, we should strengthen and foster pro-social motives in the population.

## 6. Research Limitations

It needs to be emphasized that the limitations of the present research should be borne in mind; it is possible that the results are a consequence of the small size of the research group, the method of data collection, and the fact that the research participants were not wholly representative of Polish society. Two limitations of our research seem particularly important. Firstly, we realize that in order to ensure full representativeness of the population, there should be more male participants. Nevertheless, we can see that other researchers face a similar difficulty (e.g., Prati (81.5% women) [54]; Talarowska et al. (78.6% women) [55], or Babicki et al. (83% women) [16]). Women are probably more likely to respond to requests to participate in tedious and time-consuming research. Secondly, we are aware that in Poland less than 30% of people 60+ use the internet. This indicator is two times lower than for the EU-28. In our research, this age group is the least numerous. Yet, in recent years, there has been a steady increase in the number of seniors who actively spend time in virtual reality and are fluent in using digital devices. Additionally, younger seniors differ in their functioning from older ones. However, there is a strong stereotype regarding the functioning of the elderly in the world of modern technologies (seniors are afraid, do not understand and do not need to use IT). Inviting seniors to take part in research such as ours is, in our opinion, a way to overcome this stereotype. The challenge for our future research is to motivate men and seniors to participate.

## Figures and Tables

**Table 1 ijerph-19-09507-t001:** Characteristics of study participants.

DEMOGRAPHIC VARIABLES		*n*	%
AGE	Under 22	129	30.94
	23–34 years of age	96	23.02
	35–60 years of age	174	41.73
	Over 60 years of age	18	4.32
SEX	women	372	89.21
	men	45	10.79
MARITAL STATUS	single	238	57.07
	married	143	34.29
	divorced or separated	22	5.28
	widow/widower	5	1.2
	clergy	9	2.1
CHILDREN	no	246	58.99
	yes	171	41.01
PLACE OF RESIDENCE	big city	147	35.25
	medium-size city	69	16.55
	small town	60	14.39
	village	141	33.81
EDUCATION	basic vocational	7	1.68
	secondary	44	10.55
	higher	200	47.96
	I am still learning/studying	166	39.81
EMPLOYMENT	I work full-time	184	44.12
	I work part-time	14	3.36
	I do not work	18	4.3
	pensioner	13	3.12
	pupil/student	85	20.38
	I am learning/studying and working	103	24.70

**Table 2 ijerph-19-09507-t002:** Differences between groups on sociodemographic variables.

PARAMETER		GROUP				*p*
		GROUP A (N = 104)	GROUP B (N = 92)	GROUP C (N = 221)	TOTAL (N = 417)	
AGE [YEARS]	MSD	37.98 ± 1 5.06	32.13 ± 1 3.14	33.62 ± 1 3.37	34.38 ± 1 3.9	*p* = 0.006 *
	median	3 7.5	24	29	31	
	quartiles	23–50	21–4 1.5	22–45	22–45	A > C,B
SEX	Women	91 (8 7.50%)	88 (9 5.65%)	193 (8 7.33%)	372 (8 9.21%)	*p* = 0.078
	Men	13 (1 2.50%)	4 (4.35%)	28 (1 2.67%)	45 (10.79%)	
MARITAL STATUS	Single	52 (50.00%)	55 (59.78%)	131 (59.28%)	238 (57.07%)	*p* = 0.801
	Married	42 (40.38%)	28 (30.43%)	73 (33.03%)	143 (34.29%)	
	Divorced/Separated	6 (5.77%)	6 (6.52%)	10 (4.52%)	22 (5.28%)	
	Widow/Widower	2 (1.92%)	1 (1.09%)	2 (0.90%)	5 (1.20%)	
	Clergy	2 (1.92%)	2 (2.17%)	5 (2.26%)	9 (2.16%)	
CHILDREN	No	55 (52.88%)	56 (60.87%)	135 (61.09%)	246 (58.99%)	*p* = 0.343
	Yes	49 (47.12%)	36 (39.13%)	86 (38.91%)	171 (41.01%)	
PLACE OF RESIDENCE	Big city	44 (42.31%)	28 (30.43%)	75 (33.94%)	147 (35.25%)	*p* = 0.479
	Medium-sized city	17 (16.35%)	14 (15.22%)	38 (17.19%)	69 (16.55%)	
	Small town	16 (15.38%)	13 (14.13%)	31 (14.03%)	60 (14.39%)	
	Village	27 (25.96%)	37 (40.22%)	77 (34.84%)	141 (33.81%)	
EDUCATION	Primary/secondary	0 (0.00%)	0 (0.00%)	0 (0.00%)	0 (0.00%)	*p* = 0.082
	Basic vocational	1 (0.96%)	2 (2.17%)	4 (1.81%)	7 (1.68%)	
	Average	6 (5.77%)	14 (1 5.22%)	24 (10.86%)	44 (10.55%)	
	Higher	62 (5 9.62%)	36 (3 9.13%)	102 (4 6.15%)	200 (4 7.96%)	
	I am still learning/studying	35 (3 3.65%)	40 (4 3.48%)	91 (4 1.18%)	166 (3 9.81%)	
EMPLOYMENT	I work full-time	52 (50.00%)	28 (30.43%)	104 (47.06%)	184 (44.12%)	*p* = 0.004 *
	I work on a casual basis	2 (1.92%)	8 (8.70%)	4 (1.81%)	14 (3.36%)	
	I am not working	5 (4.81%)	6 (6.52%)	7 (3.17%)	18 (4.32%)	
	I am a pensioner	7 (6.73%)	3 (3.26%)	3 (1.36%)	13 (3.12%)	
	Pupil/student	18 (1 7.31%)	21 (2 2.83%)	46 (20.81%)	85 (20.38%)	
	I am learning/studying and working	20 (1 9.23%)	26 (2 8.26%)	57 (2 5.79%)	103 (2 4.70%)	

*p*—for quantitative variables Kruskal–Wallis test + post-hoc analysis (Dunn’s test), for qualitative variables chi-square test or Fisher’s exact test. * Statistically significant difference (*p* < 0.05). Group A—people worried about insufficient vaccination of the population, Group B—people worried about mandatory vaccination, Group C—people who do not declare concerns about vaccination.

**Table 3 ijerph-19-09507-t003:** Differences between groups on COVID-19 variables, economic situation, and sources of stress.

PARAMETER		GROUP				*p*
		GROUP A (N = 104)	GROUP B (N = 92)	GROUP C (N = 221)	TOTAL (N = 417)	
Economic conditions	My standard of living has not changed	57 (54.81%)	40 (43.48%)	104 (47.06%)	201 (48.20%)	*p* = 0.53
	My standard of living has decreased	31 (29.81%)	31 (33.70%)	73 (33.03%)	135 (32.37%)	
	My standard of living has risen	16 (15.38%)	21 (22.83%)	44 (19.91%)	81 (19.42%)	
Have you had COVID-19?	Not	66 (63.46%)	52 (56.52%)	134 (60.63%)	252 (60.43%)	*p* = 0.609
	Yes	38 (36.54%)	40 (43.48%)	87 (3 9.37%)	165 (39.57%)	
Has anyone in your family had COVID-19?	Not	31 (29.81%)	34 (36.96%)	95 (4 2.99%)	160 (38.37%)	*p* = 0.071
	Yes	73 (70.19%)	58 (63.04%)	126 (5 7.01%)	257 (61.63%)	
Which of the following situations stress you the most? **	Lack of social contacts	34 (32.69%)	33 (35.87%)	74 (33.48%)	141 (33.81%)	*p* = 0.886
	Lack of respirators and medical staff in hospitals	37 (35.58%)	22 (23.91%)	56 (25.34%)	115 (27.58%)	*p* = 0.105
	Unemployment or the prospect of losing one’s job	18 (17.31%)	17 (18.48%)	48 (21.72%)	83 (19.90%)	*p* = 0.602
	Possibility of contracting COVID-19	31 (29.81%)	12 (13.04%)	51 (23.08%)	94 (22.54%)	*p* = 0.019 *
	The possibility of my loved ones contracting COVID-19	70 (67.31%)	33 (35.87%)	105 (47.51%)	208 (49.88%)	*p* < 0.001 *
	Online teaching	44 (42.31%)	23 (25.00%)	62 (28.05%)	129 (30.94%)	*p* = 0.013 *
	Restrictions	25 (24.04%)	46 (50.00%)	65 (29.41%)	136 (32.61%)	*p* < 0.001 *
	My family’s financial problems	22 (21.15%)	23 (25.00%)	50 (2 2.62%)	95 (22.78%)	*p* = 0.812
	National economic situation	72 (69.23%)	53 (5 7.61%)	111 (50.23%)	236 (56.59%)	*p* = 0.005 *
	Political situation in the country	84 (80.77%)	51 (5 5.43%)	127 (57.47%)	262 (62.83%)	*p* < 0.001 *
	Impeded access to treatment for other diseases	66 (63.46%)	60 (6 5.22%)	131 (59.28%)	257 (61.63%)	*p* = 0.558
	Other	4 (3.85%)	8 (8.70%)	11 (4.98%)	23 (5.52%)	*p* = 0.292

*p*—for quantitative variables Kruskal–Wallis test, for qualitative variables chi-square test or Fisher’s exact test. * Statistically significant difference (*p* < 0.05); ** Multiple choice question—percentages do not add up to 100. Group A—people worried about insufficient vaccination of the population, Group B—people worried about mandatory vaccination, Group C—people who do not declare concerns about vaccination.

**Table 4 ijerph-19-09507-t004:** Differences between groups on psychological variables.

PARAMETER			GROUP			
			GROUP A (N = 104)	GROUP B (N = 92)	GROUP C (N = 221)	*p*
UMACL	HT	M/SD	25.16 ± 2.01	24.89 ± 1.82	25.21 ± 2.03	*p* = 0.387
		median	25	25	25	
		quartiles	24–26	24–26	24–26	
	TA	M/SD	22.62 ± 3.86	22.12 ± 3.68	21.94 ± 3.67	*p* = 0.262
		median	23	22	21	
		quartiles	20–26	20–25	19–25	
	EA	M/SD	23.3 ± 3.91	23.98 ± 4.05	23.91 ± 3.7	*p* = 0.17
		median	23	2 3.5	24	
		quartiles	20–26	21–27	22–26	
PANAS	PA	M/SD	28.05 ± 8.38	25.45 ± 9.34	26.3 ± 7.97	*p* = 0.071
		median	29	25	27	
		quartiles	23.75–33	17–32	20–31	
	NA	M/SD	22.89 ± 9.27	21.55 ± 9.12	20.37 ± 8.43	*p* = 0.072
		median	2 1.5	20	18	
		quartiles	1 4.75–30	14–2 8.25	14–25	
SWLS		M/SD	20.28 ± 6.83	19.33 ± 7.01	19.55 ± 5.91	*p* = 0.373
		median	21	18	20	
		quartiles	15–26	14–25	15–24	
LOT-R		M/SD	14.55 ± 4.8	14.23 ± 5.48	14.18 ± 4.5	*p* = 0.836
		median	15	15	15	
		quartiles	11–18	11–18	11–17	
CISS	TOC	M/SD	58.07 ± 8.92	55.67 ± 9	56.55 ± 8.57	*p* = 0.151
		median	58	55	56	
		quartiles	51–65	50–63	51–62	
	EOC	M/SD	47.66 ± 12.02	49.66 ± 13.13	47.64 ± 10.59	*p* = 0.241
		median	47	51	47	
		quartiles	39–5 6.25	3 8.75–61	40–55	
	AOC	M/SD	46.65 ± 8.12	47.65 ± 9.57	47.27 ± 8.5	*p* = 0.8
		median	48	47	48	
		quartiles	40–52	41.75–54	41–54	
	D	M/SD	20.56 ± 5.04	2 1.82 ± 6.08	21.53 ± 5.43	*p* = 0.224
		median	21	2 1.5	22	
		quartiles	17–24	18–25	17–26	
	SD	M/SD	17.76 ± 3.76	17.16 ± 4.03	17.35 ± 3.88	*p* = 0.474
		median	18	17	17	
		quartiles	15–20	14–20	15–20	

*p*—Kruskal–Wallis test. Group A—people worried about insufficient vaccination of the population, Group B—people worried about mandatory vaccination, Group C—people who do not declare concerns about vaccination.

## Data Availability

The data presented in this study are available on request from the corresponding author.

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
