# Peer review of "Sociodemographic and Psychological Variables and Concerns Related to COVID-19 Vaccination among Polish Citizens"

_ijerph, 2022, doi:10.3390/ijerph19159507_

Round 1

Reviewer 1 Report

The authors attempted to investigate if there are any differences in socio-demographic and physiological variables for a surveyed Polish population categorized into three groups: (1) people who worry about insufficient vaccination of the entire population, (2) people who worry about mandatory vaccination, and (3) people who do not declare concerns about vaccination. To achieve it, they designed an online survey questionnaire that was filled in Google Form and the questionnaire was distributed through personal contacts and social media. Using the study population, they demonstrated that elderly and working people are more worried about insufficient vaccination. On the contrary, students are more concerned about mandatory vaccination. This current version cannot be accepted without a major revision. My major and minor comments are as follows:

Major issues:

  1. Co-founding factors are not mentioned in the paper. Are co-founding factors or covariates collected in the survey? For example, as the authors mentioned in the introduction, factors such as the perceived risk of becoming infected with COVID-19 can influence their results. What is the authors’ approach to solving the existence of those co-founding variables?
  2. Only the mean and standard deviation of ages for people in the study are mentioned. However, this description is way too coarse. I would appreciate it if an age distribution can be shown. 
  3. Related to the previous comment (i.e. point 2), I am worried that there might be a bias in the age stratification due to the design of the digital questionnaire. For example, it is likely that some elderly participants may not be good at handling digital survey forms. Hence, the age distribution might bias toward a younger population.
  4. The authors briefly talked about how their results could be potentially applied to real-life scenarios. I would appreciate more discussions about how their findings can inform the policy-making for boosting the vaccination rate.

Minor comments:

  1. Lines 25-26: “more than 6,301,000 people worldwide have died from COVID-25 19, including 116,268 in Poland…”. Authors need to specify when those numbers were obtained (i.e. until which date). The numbers change over time, the time range has to be specified.
  2. Lines 61-62: “It is estimated that vaccinating 60-72% of the population can lead to so-called herd immunity…”. Authors need to point out that the estimation is performed for which SARS-COV-2 variant and what R0 is assumed. That estimation is very likely to fail on new variants such Omicron variant. The authors need to clarify this.

Author Response

Dear Reviewer,

Thank you very much for your meaningful comments. We have tried to address all your suggestions and we believe that this helped us to improve the new version of the paper. Please find our detailed response below.

Comments and Suggestions for Authors

The authors attempted to investigate if there are any differences in socio-demographic and physiological variables for a surveyed Polish population categorized into three groups: (1) people who worry about insufficient vaccination of the entire population, (2) people who worry about mandatory vaccination, and (3) people who do not declare concerns about vaccination. To achieve it, they designed an online survey questionnaire that was filled in Google Form and the questionnaire was distributed through personal contacts and social media. Using the study population, they demonstrated that elderly and working people are more worried about insufficient vaccination. On the contrary, students are more concerned about mandatory vaccination. This current version cannot be accepted without a major revision. My major and minor comments are as follows:

Major issues:

Co-founding factors are not mentioned in the paper. Are co-founding factors or covariates collected in the survey? For example, as the authors mentioned in the introduction, factors such as the perceived risk of becoming infected with COVID-19 can influence their results. What is the authors’ approach to solving the existence of those co-founding variables?

Thank you for this comment. We agree that the perceived risk of infection with COVID-19 is an important factor influencing the acceptance of vaccination. As the Reviewer kindly noted, we mentioned about it in the introduction. Nevertheless, we did not want to copy the research assumptions of other authors, so in our research we focused not on the cognitive aspect of the attitude towards vaccination, but on the emotional one. Therefore, we examined various concerns, including these COVID-19 and vaccination-related.

Nevertheless, the results obtained by other authors in terms of perceived risk of infection interestingly correspond to data showing that people who are concerned about their own and their loved ones' COVID-19 infection are also concerned about insufficient vaccination of the population (see: Table 3).

Only the mean and standard deviation of ages for people in the study are mentioned. However, this description is way too coarse. I would appreciate it if an age distribution can be shown.

Thank you for this comment. We have completed the data on age distribution in Table 1.

Related to the previous comment (i.e. point 2), I am worried that there might be a bias in the age stratification due to the design of the digital questionnaire. For example, it is likely that some elderly participants may not be good at handling digital survey forms. Hence, the age distribution might bias toward a younger population.

We strongly agree with this comment. To emphasize its importance, we have separated the part of the article devoted to the limitations of our research.

We are aware that in Poland, according to the data of the Central Statistical Office (GUS, 2019), less than 30% of people 60+ uses the Internet. This indicator is two times lower than for the EU-28. In our research, this group is the least numerous. And yet, in recent years, there has been a steady increase in the number of seniors who actively spend time in virtual reality and are fluent in using digital devices. Additionally - younger seniors differ in their functioning from older ones. However, there is a strong stereotype regarding the functioning of the elderly in the world of modern technologies (seniors are afraid, do not understand and do not need to use IT). Inviting seniors to take part in research such as ours is, in our opinion, a way to overcome this stereotype. We agree that reaching the group of older seniors is not easy - so we used not only the tools of new technologies, but also personal contacts (with moderate success).

The authors briefly talked about how their results could be potentially applied to real-life scenarios. I would appreciate more discussions about how their findings can inform the policy-making for boosting the vaccination rate.

Thank you for your comment. We supplemented the discussion with practical implications to promote the National COVID-19 Vaccination Program.

Thanks to scientific research conducted almost all over the world, we have an increasing knowledge of the factors underlying attitudes towards vaccination against COVID-19. The conducted research focuses on various aspects of these attitudes: cognitive, psychological, demographic, political and cultural.

In our research, we focused on demographic variables and, what we consider to be an added value of our research, on psychological variables. Our study showed that there are differences between people who are concerned about the inadequate vaccination in the population and those who are concerned about mandatory vaccinations. It is worth nothing that these differences concern only demographic and not psychological variables measured with standardized psychological tools.

This knowledge makes it possible to design campaigns promoting vaccination against COVID-19 aimed at specific target groups, which are easy to identify taking into account the demographic data available in our country.

Following this, efforts to promote vaccination against COVID-19 should target young people, learning and studying. With this in mind, campaigns should be conducted in schools and universities, and should be publicized through social media, i.e. in places where young people spend their time. In addition, vaccination campaigns should involve young, popular people with whom young people can identify.

But not only. It turns out that people who are concerned about insufficient vaccination of the population are also concerned about the health of their loved ones. The awakening of motivation focused on love, care and responsibility seems to be a factor enhancing the readiness to vaccinate (Lamy et al. 2022). Therefore, campaigns should also be aimed at strengthening pro-social motives.

Minor comments:

Lines 25-26: “more than 6,301,000 people worldwide have died from COVID-25 19, including 116,268 in Poland…”. Authors need to specify when those numbers were obtained (i.e. until which date). The numbers change over time, the time range has to be specified.

Thank you for this comment. We updated the data and added the access date “6 387 770; 116 470 (18.07.2022)”).

Lines 61-62: “It is estimated that vaccinating 60-72% of the population can lead to so-called herd immunity…”. Authors need to point out that the estimation is performed for which SARS-COV-2 variant and what R0 is assumed. That estimation is very likely to fail on new variants such Omicron variant. The authors need to clarify this.

Thank you for your comment. We supplemented the text with your meaningful observation.

(“However, it should be borne in mind that the SARS-Cov-2 coronavirus continues to mutate and it is very likely that the estimation will fail for novel coronavirus variants”).

Thank you once again for your suggestions, we hope that we have managed to remove all identified shortcomings.

Kind regards,

Authors

Reviewer 2 Report

This manuscript describe sociodemographic and psychological variables and concerns related to COVID-19 vaccination among Polish citizens, aiming to identify risk groups in order to adapt social campaigns promoting vaccination against COVID-19 in Poland. However, the result can’t support this purpose.

1. The dependent variable is gained by a question about factors that contributed to their stress in the current situation and is categorized as concerned about inadequate vaccination of the population (A), concerned about the prospect of mandatory vaccination (Group B) and no concerns about vaccination (Group C). How to promote vaccination by the factors influencing this variable? Therefore the dependent variable needs to be reconsidered.

2. I don’t think it is proper to analyse the association of psychological variables (Mood, Emotions, Satisfaction with life Optimism, Coping with stress) and  the dependent variable of this study.

3.  Most of the participants were female. The selection bias is out of that can be acceptable.

4. Question of “Which of the following situations stress you the most?” is a meaningful variable to promote vaccination and other intervention approach. I recommend the authors can analyze who and how many people having these stress. The results of this might be helpful for vaccination promotion.

5. The introduction is rigmarole.

6. Line 128-130, analyse the differences in sociodemographic and psychological variables between those who cite fears related to vaccination and those who do not report such fears, as well as the differences between those whose fears result from a negative evaluation of the vaccines and those whose fears result from a positive evaluation.  The statement doesnt accord with the dependent variable analyzed in the result part. Whats the relationship between fears related to vaccination, fears result from a negative (positive) evaluation of the vaccines and “factors that contributed to their stress in the current situation”?

Author Response

Dear Reviewer,

Thank you very much for your meaningful comments. We have tried to address all your suggestions and we believe that this helped us to improve the new version of the paper. Please find our detailed response below.

This manuscript describe sociodemographic and psychological variables and concerns related to COVID-19 vaccination among Polish citizens, aiming to identify risk groups in order to adapt social campaigns promoting vaccination against COVID-19 in Poland. However, the result can’t support this purpose.

  1. The dependent variable is gained by a question about factors that contributed to their stress in the current situation and is categorized as concerned about inadequate vaccination of the population (A), concerned about the prospect of mandatory vaccination (Group B) and no concerns about vaccination (Group C). How to promote vaccination by the factors influencing this variable? Therefore the dependent variable needs to be reconsidered.

Thank you for this comment. We probably did not clearly define the purpose of our research, as the aim of the study was to analyse the differences in sociodemographic and psychological variables between those concerned about inadequate vaccination of the population, those concerned about the prospect of mandatory vaccination, and those who do not indicate concerns about vaccination.

In our research, we focused not on the cognitive aspect of attitudes, but on the emotional one. Therefore, the selection criterion for the 3 groups was the question about concerns related to vaccinations.

Moreover, the obtained research results are not intended to promote vaccinations, but only to formulate suggestions to which target groups (young people, pupils and students) the social campaign should be directed.

We have tried to clarify this part of the text so that it does not mislead the reader.

  1. I don’t think it is proper to analyse the association of psychological variables (Mood, Emotions, Satisfaction with life Optimism, Coping with stress) and the dependent variable of this study.

Thank you for this comment. We agree that the analysis of the relationship between the dependent variable and psychological variables did not show any differences between groups, but we wouldn’t be able to tell it without our research. The lack of differences is in our opinion an important information, because it shows that demographic factors play a greater role than psychological ones in terms of COVID-related concerns.

We find it particularly interesting that despite the greater number of stressors experienced by people who are afraid of insufficient vaccination of the population, we do not observe lower scores on the emotional well-being scales, while the logical consequence of a greater number of stressors would be a deterioration of e.g. mood. Perhaps the very attitude towards vaccination (we tentatively assume that people who are afraid of insufficient vaccination of the population accept and are more likely to vaccinate) is a protective factor. However, this is, of course, only a hypothesis that requires further verification.

  1. Most of the participants were female. The selection bias is out of that can be acceptable.

We strongly agree with this comment. To emphasize its importance, we have separated the part of the article devoted to the limitations of our research.

We realize that in order to ensure full representativeness of the population there should be more male participants. Nevertheless, we can see that in many of the studies we cite, there are many more women than men (e.g. Prati, 2021 (81.5% women); Talarowska et al. al., 2020 (78.6% women) or Babicki et al, 2021 (83% women)). Women are probably more likely to respond to requests to participate in tedious and time-consuming research. We realize that the challenge for our future research is to motivate men to participate.

  1. Question of “Which of the following situations stress you the most?” is a meaningful variable to promote vaccination and other intervention approach. I recommend the authors can analyze who and how many people having these stress. The results of this might be helpful for vaccination promotion.

Thank you for your valuable comment. We have added a paragraph in which, according to your suggestion, we show how many people from each of the groups (and in total sample) are indicating particular stressors.

We have also added some topics in the discussion inspired by your suggestion.

  1. The introduction is rigmarole.

Thank you for this comment. It allowed us to take another look at the introduction. To emphasize its structure, it has been divided into parts. The part at the end of the introduction has also been expanded to explain more clearly the idea of our research and the choice of the dependent variable.

  1. Line 128-130, “analyse the differences in sociodemographic and psychological variables between those who cite fears related to vaccination and those who do not report such fears, as well as the differences between those whose fears result from a negative evaluation of the vaccines and those whose fears result from a positive evaluation”. The statement doesn’t accord with the dependent variable analyzed in the result part. What’s the relationship between “fears related to vaccination”, “fears result from a negative (positive) evaluation of the vaccines” and “factors that contributed to their stress in the current situation”?

Our study focuses on the emotional aspect of attitudes towards vaccination, with particular emphasis on concerns about vaccination. They may take the form of concerns about the consequences of inadequate vaccination of the population (we assume that such concerns occur in people who positively assess vaccination) and the form of concerns about the consequences of mandatory vaccination (we assume that such concerns occur in people who negatively assess vaccination). We also assume that in a certain part of the population none of these are present.

We analysed the differences between the 3 groups in terms of sociodemographic and psychological variables: the first group were people who declared that they were concerned about insufficient vaccination of the population (Group A) n = 104 (24.94%), the second group were people who declared that they were concerned about the prospect of mandatory vaccination (Group B) n = 92 (22.06%) and the third group of people who do not report vaccine concerns (Group C) n = 221 (53%).

Following your interesting suggestion, we have also expanded the analysis of the differences between these groups in terms of experiencing other concerns.

Thank you once again for your suggestions, we hope that we have managed to remove all identified shortcomings.

Kind regards,

Authors

Round 2

Reviewer 1 Report

The authors answered the questions I raised. I have no further questions.

Reviewer 2 Report

Thanks for effort authors has made for revision. The current version is fine for publish.